# Microbial Transformation of Pimavanserin by *Cunninghamella blakesleeana* AS 3.970

Ming Song [1], Qi Yu [2], Yuqi Liu [1], Sulan Cai [1], Xuliang Jiang [3], Weizhuo Xu [1,*] and Wei Xu [1,*]

[1] School of Functional Food and Wine, Shenyang Pharmaceutical University, 103 Wenhua Road, Shenhe District, Shenyang 110016, China; songming024@163.com (M.S.); liuyq1019@163.com (Y.L.); cai-sulan@163.com (S.C.)

[2] Heilongjiang Institute for Drug Control, Wanggang Street 711, Harbin 150088, China; yq86330695@163.com

[3] School of Pharmaceutical Engineering, Shenyang Pharmaceutical University, 103 Wenhua Road, Shenhe District, Shenyang 110016, China; xuliangjiang@syphu.edu.cn

* Correspondence: weizhuo.xu@syphu.edu.cn (W.X.); shxuwei8720@163.com (W.X.); Tel./Fax: +86-024-43520301 (Weizhuo Xu); +86-024-43520307 (Wei Xu)

**Abstract:** Pimavanserin is an approved selective 5-HT$_{2A}$ receptor inverse agonist for treating Parkinson's disease psychosis. However, few studies on its metabolism in vitro have been investigated. In this research, eight strains of fungi are used to study the pimavanserin metabolism profiles in vitro and six of them demonstrated positive transformation results. Factors influencing the transformation rate, like substrate concentration, culture time, initial media pH value, culture temperature, and shaking speed, were evaluated and optimized. *Cunninghamella blakesleeana* AS3.970 provided the best transformation rate of 30.31%, and 10 unreported metabolites were screened by LC-MS/MS. Among these metabolites, M1 is the major one and identified as 1-(4-fluorobenzyl)-3-(4-(2-hydroxy-2-methylpropoxy)benzyl)-1-(1-methylpiperidin-4-yl)urea, which is a hydroxylation product of the pimavanserin. A preliminary molecular docking simulation was performed, which indicated that M1 exhibits similar binding properties with pimavanserin and may become a potential candidate for Parkinson's disease treatment.

**Keywords:** biocatalysis; pimavanserin; fungi; *Cunninghamella blakesleeana*

## 1. Introduction

Pimavanserin is a selective 5-HT$_{2A}$ receptor inverse agonist, which has the IUPAC name of *N*-(4-fluorophenylmethyl)-*N*-(1-methylpiperidin-4-yl)-*N'*-(4-(2-methylpropyloxy) phenylmethyl)carbamide. It is the first FDA-approved drug for the treatment of hallucinations and delusions associated with Parkinson's disease [1]. In the molecule, three of the four hydrogens are replaced by 4-fluorobenzyl, 1-methylpiperidin-4-yl, and 4-(isopropyloxy) benzyl groups, which made it a selective and potent serotonin 2A (5-HT$_{2A}$) receptor and an inverse agonist and antagonist [2]. Pimavanserin had a lower affinity in membranes and whole cells and lacked affinity and functional activity at the 5-HT$_{2B}$ receptors and dopamine D2 receptors [3].

Drug metabolism paves the way for the fundamental development of new drug entities to be further evaluated for pharmacological/toxicological activities. The formation of metabolite and its role in the body before excretion is important to understand the drug's safety and toxicity profile [4]. Metabolism is defined as the structural modification of drugs and chemicals by enzymatic systems which lead to the formation of relatively polar substances that are easily excreted from the organism [5]. Many fungi possess cytochrome P-450 monooxygenase systems analogous to those in mammals, so microbial transformation has been proposed as a model system for mammalian drug metabolism [6].

*Cunninghamella* is a genus of fungus belonging to family Cunninghamellaceae which can be found in soil, plants, and animals [7]. *Cunninghamella* has properties to metabolize a variety of drugs, more than a hundred of them, in a way similar to the mammalian enzyme

systems [8]. Among those multiple enzyme systems, cytochrome P450s represent the most important class of enzymes involved in phase I metabolism, which is involved in 75–80% of the metabolism of marketed drugs. Phase I reactions involve various functionalization reactions, such hydroxylation, epimerization, hydrolysis, oxidation, and reduction [9,10]. Hence, as an alternative of chemical route to generate and derivate pharmacologically active compounds in vivo, microbial transformation and conversion of drugs have been properly implemented.

To evaluate the microbial metabolism of pimavanserin, eight strains of *Cunninghamella* and other species fungi were selected for transformation. From the transformation solutions, 10 pimavanserin-derived metabolites were screened, in which one of the hydroxylation derivative, 1-(4-fluorobenzyl)-3-(4-(2-hydroxy-2-methylpropoxy)benzyl)-1-(1-methylpiperidin-4-yl) urea was prepared and identified. A preliminary molecular docking was simulated between this hydroxylation product and pimavanserin to compare their combination ability with the 5-HT$_{2A}$ receptors.

## 2. Results and Discussion

### 2.1. Identification of Efficient Transformation Microorganism Strains

The HPLC results indicated that six strains exhibited the positive transformation metabolites. The pimavanserin transformation rate could be calculated according to the peak area normalization method, i.e., the conversion rate (%) = $A_{product}/(A_{product} + A_{substrate})$.

*C. blakesleeana* AS 3.970, *C. blakesleeana* AS 3.910, *C. blakesleeana* AS 3.153, *C. elegans* AS 3.156 and *C. elegans* AS 3.2028 displayed the abilities to transform pimavanserin in both growth cell and resting cell transformation. *M. cyclinelloides* AS 3.3421 could transform pimavanserin by growth cell transformation only. *A. niger* AS 3.739 and *E. cristatum* AS 3.793 could not transform pimavanserin at all. The results are summarized in Table 1. Among all these strains, *C. blakesleeana* AS 3.970 showed the highest product yield of 16.24% by resting cell transformation (Figure 1).

**Table 1.** Results of pimavanserin microbial transformation by filamentous fungi.

| Fungus Species | Resting Cell Transformation Rate (%) | Growth Cell Transformation Rate (%) |
| --- | --- | --- |
| *Cunninghamella blakesleeana* AS 3.970 | 16.24 | 7.10 |
| *Cunninghamella blakesleeana* AS 3.910 | 10.46 | 4.64 |
| *Cunninghamella blakesleeana* AS 3.153 | 7.68 | 3.12 |
| *Cunninghamella elegans* AS3.156 | 7.42 | 2.44 |
| *Cunninghamella elegans* AS3.2028 | 4.58 | 1.12 |
| *Mucor circinelloides* AS 3.3421 | 2.32 | - |
| *Aspergillus niger* AS 3.739 | - | - |
| *Eurotium cristatum* AS 3.793 | - | - |

### 2.2. Optimization of C. blakesleeana AS 3.970 Resting Cell Biotransformation

As the *C. blakesleeana AS 3.970* demonstrated the best transformation rate of pimavanserin, it is beneficial to optimize the biotransformation process. From Figure 2A, it can be seen that as the pimavanserin concentration increases, the specific transformation rate gradually decreases. However, when the pimavanserin concentration is 0.04 mg/mL, the absolute mass of the product is obtained (absolute mass = pimavanserin mass × transformation rate) is the highest, therefore, the optimal concentration for the substrate is 0.04 mg/mL. As the culture time increases, the transformation rate increases and tends to stabilize at 72 h (Figure 2B). The pH value of the culture medium is closely related to microbial life activities, which could affect the microorganism growth, metabolites accumulation, nutrients utilization, as well as the structure and activity of enzymes. When the initial pH is below 6.5, the transformation rate gradually increases. When the initial pH increases again, the transformation rate gradually decreases. Therefore, the optimal pH is 6.5 (Figure 2C). The results in Figure 2D show that in the process of microbial transformation, it is necessary

to maintain appropriate temperature, which can affect the rate of enzyme catalysis to ensure the normal growth of bacteria and the synthesis of transformation products. The shaking speed has a significant impact on the growth of mycelium. The appropriate shaking speed not only meets the demand for dissolved oxygen in the mycelium, but also helps the mycelium not to clump. If the shaking speed is too low, it will cause insufficient dissolved oxygen and mycelium nodulation. If the shaking speed is too high, it is easy to break the mycelium, which is not conducive to the growth of the mycelium and the transformation rate. The results in Figure 2E indicate that the transformation rate is highest at a speed of 180 rpm. Summarily, an optimized transformation process was set as 0.04 mg/mL substrate concentration, 72 h culture time, initial pH 6.5, 28 °C at 180 rpm, and the transformation rate increased to 30.31% (Figure 2F).

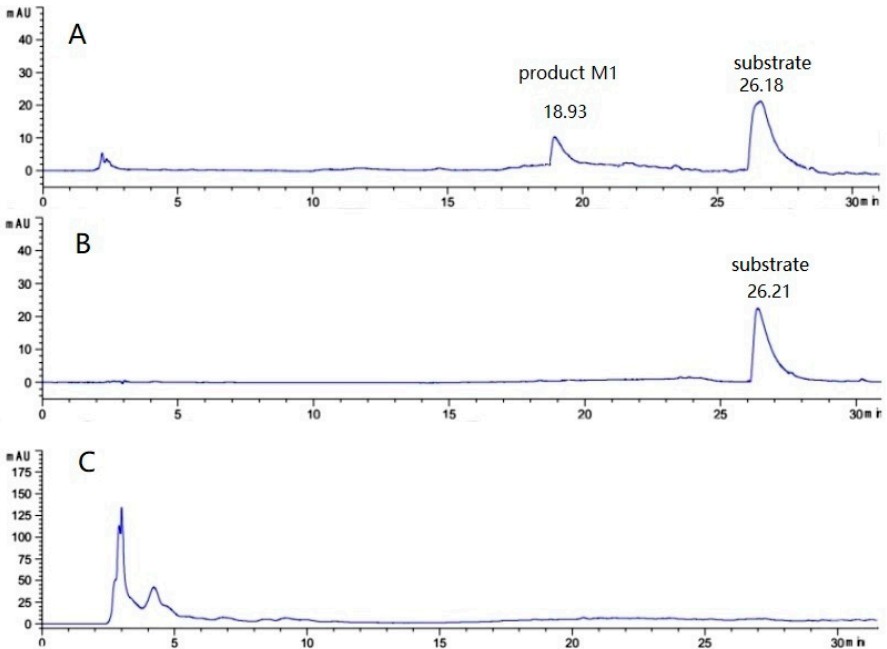

**Figure 1.** HPLC results of *C. blakesleeana* AS 3.970 resting cell transformation. (**A**). Substrate transformation group (pimavanserin + *C. blakesleeana* AS 3.970); (**B**). substrate control group (pimavanserin); (**C**). Strain control group (*C. blakesleeana* AS 3.970).

### 2.3. Preparation and Identification of Major Biotransformation Product M1

The transformation product (6.8 mg) was purified by the semi- and preparative HPLC and detected by LC-ESI-MS and NMR.

The HPLC detection results of the transformation product are shown in Figure 3A, indicating that the product obtained through separation and purification has a single component and almost no impurities. The molecular weight of the product is 444 Da (Figure 3B), which gave a 16-increment compared with pimavanserin of 428 Da, so an oxygen atom addition into pimavanserin could be identified. Its NMR spectra (Table 2) showed that the chemical shifts of the original two methyl groups at positions 24 and 25 ($\delta_C$: 19.54 ppm, $\delta_H$: 0.96 ppm) and the carbon at position 23 ($\delta_C$: 28.18 ppm) all moved significantly to the low field ($\delta_C$: 27.11 ppm; $\delta_H$: 1.23 ppm and $\delta_C$: 69.11 ppm, respectively), and the hydrogen on the carbon at position 23 disappeared. These changes are consistent with the characteristics of carbon at position 23 being substituted by a hydroxyl group, and this is a compound has not reported before. (M1, Figure 3C).

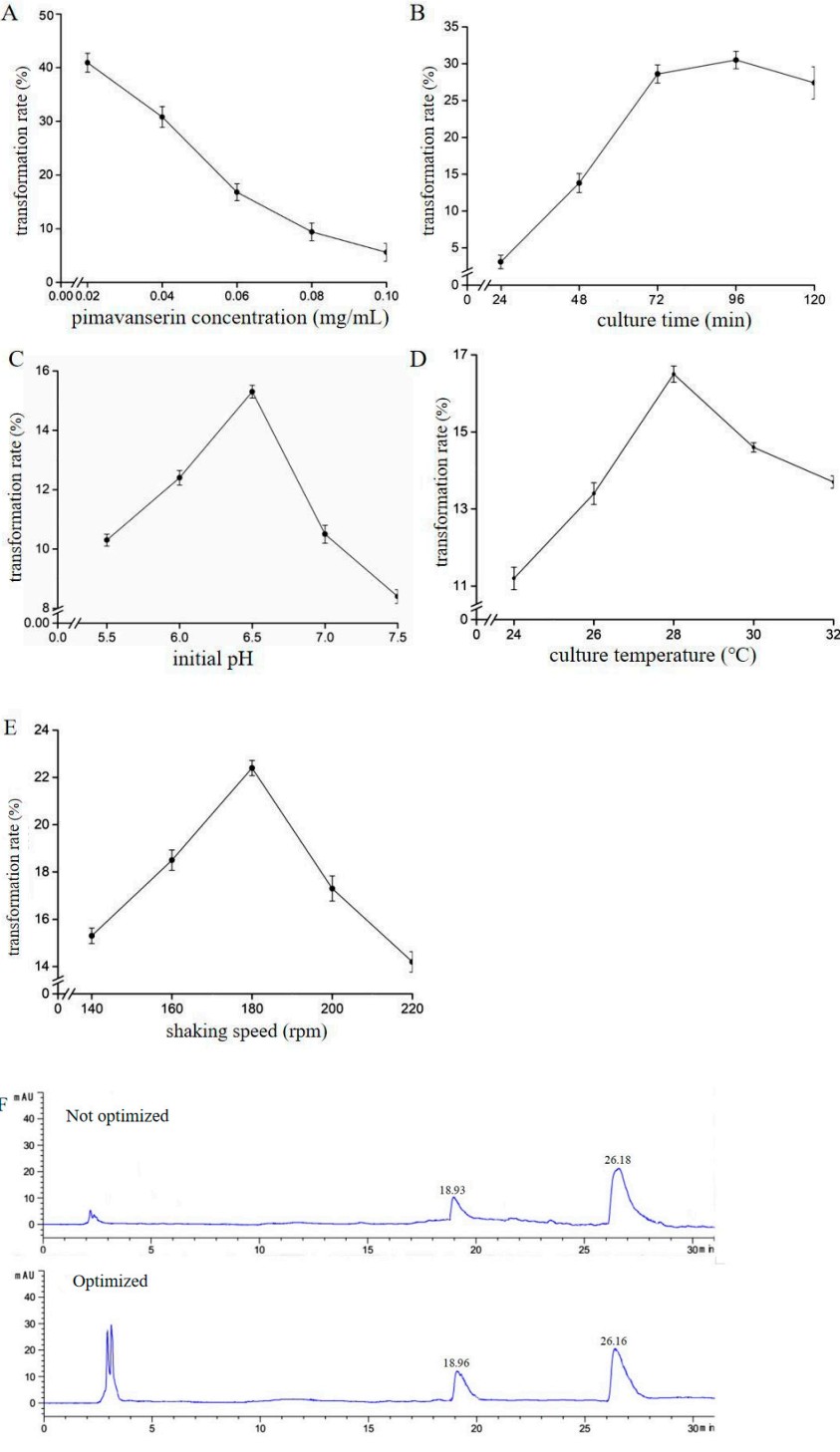

**Figure 2.** Optimization of transformation conditions. (**A**). 5% inoculation volume, 100 mL medium, 28 °C, initial pH 6.5, 200 rpm, and culture time 48 h; (**B**). 5% inoculation volume, 100 mL medium, 28 °C, initial pH 6.5, 200 rpm, and pimavanserin concentration 0.06 mg/mL; (**C**). 5% inoculation volume, 100 mL medium, 28 °C, 200 rpm, pimavanserin concentration 0.06 mg/mL, and culture time 48 h; (**D**). 5% inoculation volume, 100 mL medium, pH 6.5, 200 rpm, pimavanserin concentration 0.06 mg/mL, and culture time 48 h; (**E**). 5% inoculation volume, 100 mL medium, 28 °C, pH 6.5, pimavanserin concentration 0.06 mg/mL, and culture time 48 h; (**F**). comparison of HPLC before and after condition optimization.

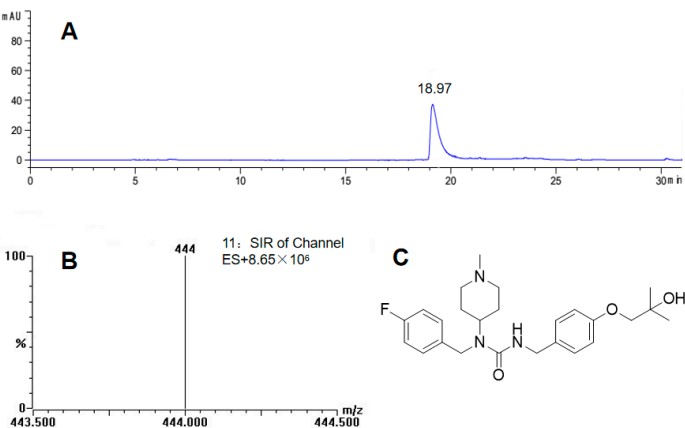

**Figure 3.** LC-ESI-MS analysis of products. (**A**). HPLC diagram of metabolite M1 by preparative HPLC. (**B**). ESI-MS diagram of metabolite M1. (**C**). Resolved structure of Metabolite M1.

**Table 2.** The $^1$H NMR and $^{13}$C NMR spectra data for the transformation product (M1) [a,b].

| Position | $\delta_C$ | $\delta_H$ |
|---|---|---|
| 1 | 162.19 | - |
| 2 | 115.37 | 7.05 (1H, m) |
| 3 | 128.84 | 7.25 (1H, m) |
| 4 | 137.19 | - |
| 5 | 128.79 | 7.25 (1H, m) |
| 6 | 115.37 | 7.05 (1H, m) |
| 7 | 51.57 | 4.93 (2H, m) |
| 8 | 54.22 | 3.74 (1H, m) |
| 9 | 28.94 | 1.79 (2H, m) |
| 10 | 44.59 | 2.34 (2H, m) |
| 11 | 43.57 | 2.34 (2H, m) |
| 12 | 28.94 | 1.77 (2H, m) |
| 13 | 40.32 | 2.33 (3H, m) |
| 14 | 160.59 | - |
| 15 | 44.59 | 4.72 (2H, m) |
| 16 | 128.61 | - |
| 17 | 133.48 | 7.24 (1H, m) |
| 18 | 114.61 | 6.63–6.95 (1H, d) |
| 19 | 157.94 | - |
| 20 | 115.52 | 6.63–6.95 (1H, d) |
| 21 | 113.48 | 7.25 (1H, m) |
| 22 | 76.64 | 4.08 (2H, m) |
| 23 | 69.11 | 1.71–1.77 (2H, d) |
| 24 | 27.11 | 1.23 (3H, d) |
| 25 | 27.11 | 1.23 (3H, d) |

[a] Data were measured DMSO, Coupling constants (*J* in Hz) are given in parentheses. [b] Recorded at 600 MH$_Z$ for compounds.

## 2.4. Analysis of Biotransformation Products by LC-MS/MS

Microbial transformation usually provides multiple metabolites, in this study, the 1-(4-fluorobenzyl)-3-(4-(2-hydroxy-2-methylpropoxy)benzyl)-1-(1-methylpiperidin-4-yl) urea is the only major product. To explore the more other transformation products, a LC-MS/MS

method is applied to validate the aforementioned pooled transformation solution. Finally, the LC-MS/MS analysis revealed 10 pimavanserin-derived metabolites, which are not reported yet (Figure 4). Detailed mass spectrometric data are listed in the Supplementary Information (Table S1, Figures S1–S21).

**Figure 4.** Microbial transformations of pimavanserin (M0) by *Cunninghamella*.

M0, the quasi-molecular ion [M + H]$^+$, was *m/z* 428 and its fragment ions *m/z* were 223, 163, and 98, respectively, which did not change its molecular weight compared with the original pimavanserin, thus proving that the compound is the prototype pimavanserin.

M2, the quasi-molecular ion [M + H]$^+$, is *m/z* 460, which is 32 more molecular weight than that of pimavanserin, and the compound may be a dihydroxylated product of pimavanserin. Its fragment ions are mainly *m/z* 98 and *m/z* 223, indicating that no structural changes occurred in the 4-fluorobenzyl and 1-methylpiperidin-4-yl fractions.

M3, the quasi-molecular ion [M + H]$^+$ of this compound, is *m/z* 372, which is 56 less molecular weight than that of pimavanserin, and this compound may be a deisobutyl product of pimavanserin. Its fragment ions are mainly *m/z* 98 and *m/z* 223, indicating no structural changes in the 4-fluorobenzyl and 1-methylpiperidin-4-yl fractions.

M4, the quasi-molecular ion [M + H]$^+$ of this compound, is *m/z* 320, which is 108 less molecular weight than that of pimavanserin, and this compound may be a defluorobenzyl product of pimavanserin. The fragment ions are mainly *m/z* 98 and *m/z* 115, and the *m/z* 115 fragment ion is 1-methylpiperidin-4-amine ion.

M5, the quasi-molecular ion [M + H]$^+$ of this compound is *m/z* 266, which is 162 less molecular weight compared to pimavanserin, and this compound may be an 4-isobutoxybenzyl missing product from pimavanserin. Its fragment ions are mainly *m/z* 98 and *m/z* 223, indicating that no structural changes occurred in the 4-fluorobenzyl and 1-methylpiperidin-4-yl fractions.

M6, the quasi-molecular ion [M + H]$^+$ is *m/z* 430, which is two more molecular weight than that of pimavanserin, and this compound may be a pimavanserin derivative with one hydroxyl introduced as well as one methyl removed (2 = 16–14). Its fragment ions are mainly *m/z* 98 and *m/z* 223, indicating that no structural changes occurred in the 4-fluorobenzyl and 1-methylpiperidin-4-yl fractions.

M7, the quasi-molecular ion [M + H]$^+$ of this compound, is *m/z* 446, which is 18 more molecular weight than that of pimavanserin. This compound may be a pimavanserin derivative with two hydroxyl groups introduced as well as one methyl removed (18 = 32 − 14). Its fragment ions are mainly *m/z* 98 and *m/z* 223, indicating no structural changes in the 4-fluorobenzyl and 1-methylpiperidin-4-yl parts.

M8, the quasi-molecular ion [M + H]$^+$ of this compound is *m/z* 442, which has 14 more molecular weight than that of pimavanserin, and this compound may be a pimavanserin derivative with one methyl substituted. Its fragment ions are mainly *m/z* 98 and *m/z* 223, indicating no structural changes in the 4-fluorobenzyl and 1-methylpiperidin-4-yl portions.

M9, the quasi-molecular ion [M + H]$^+$ of this compound is *m/z* 458, which is 30 more molecular weight than that of pimavanserin, and this compound may be a pimavanserin derivative with one methoxy substituted. Its fragment ions are mainly *m/z* 98 and *m/z* 223, indicating no structural changes in the 4-fluorobenzyl and 1-methylpiperidin-4-yl parts.

M10, the quasi-molecular ion [M + H]$^+$ of the compound is *m/z* 410, which is 18 less than the molecular weight of pimavanserin, and the compound may be a pimavanserin derivative with one fluorine removed. Its fragment ions are mainly *m/z* 98 and *m/z* 205, and the *m/z* 205 fragment ion is the 1-methylpiperidin-4-yl bound to benzyl portion.

### 2.5. Preliminary Molecular Docking of 5-HT$_{2A}$ Receptor with Pimavanserin and M1

Hydroxylation of the drugs normally increase the water solubility [11]. In this study, pimavanserin is an oral available form. It is reasonable to evaluate if its hydroxylation form could present same or better pharmacological effect.

The human 5-HT$_{2A}$ receptor sequence was selected from Uniprot P28223, which is aligned with the beta2-adrenergic receptor(β2AR). From the sequence alignment result, it is feasible to construct a 5-HT$_{2A}$ receptor model by Modeller program using β2AR (PDB: 2RH1) as template. The molecular docking results of 5-HT$_{2A}$ receptor with pimavanserin and M1 are displayed in Figure 5.

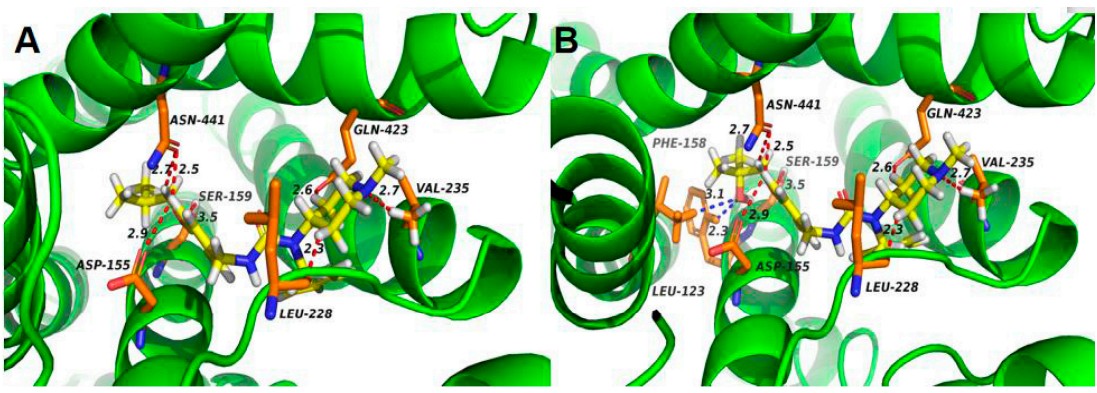

**Figure 5.** Molecular docking results of 5-HT$_{2A}$ receptor with pimavanserin (**A**) and M1 (**B**).

Compared with pimavanserin, there are two new hydrogen bonds (LEU-123, PHE158) appeared when M1 binds to the 5-HT$_{2A}$ receptor, and the binding energy changes from −9.1 kcal/mol to −9.3 kcal/mol, which makes the binding stronger. Compound M1 interacting at the lower portion of the binding site show high affinity to the receptor and the results were the same as previous literature [12], which indicates that M1 also have a good function of antagonizing 5-HT$_{2A}$ receptor and become a potential drug for Parkinson's disease treatment. So, using virtual docking technology, we investigated the binding of product M1 to the 5-HT$_{2A}$ receptor model and found that the binding of product M1 to the 5-HT$_{2A}$ receptor increased by two hydrogen bonds, resulting in a stronger binding force.

## 3. Material and Methods

### 3.1. Microorganisms and Culture Media

Those eight filamentous fungi used in this study are selected from Cunninghamella blakesleeana AS3.970, Cunninghamella blakesleeana AS 3.910, Cunninghamella blakesleeana AS 3.153, Cunninghamella elegans AS 3.2028, Cunninghamella elegans AS 3.156, Aspergillus niger AS 3.739, Eurotium cristatum AS 3.793 and Mucor circinelloides AS 3.3421, which were purchased from China General Microbiological Culture Collection Center

(CGMCC, Beijing, China), cultured and stored in our lab strain reservoir. Microbial transformation experiments were performed using the following media. Slant and solid media are consisted of peptone (10 g), NaCl (5 g), glucose (20 g), agar (15 g) and distilled $H_2O$ (1 L). The potato dextrose agar medium (PDA) are composed of peeled potato extract (200 g), glucose (20 g) and distilled $H_2O$ (1 L). The seed media is made of glucose (20 g), peptone (5 g), yeast extracts (5 g), NaCl (5 g), $K_2HPO_4$ (5 g) and distilled $H_2O$ (1 L) (pH 6.5). The bran media ingredients are peptone (10 g), NaCl (5 g), glucose (40 g), wheat bran (10 g) and distilled $H_2O$ (1 L). All culture medium was autoclaved at 121 °C for 30 min before use. P450 enzymes from above microorganisms could be found from CYtochrome P450 Engineering Database from BioCatNet (https://cyped.biocatnet.de/ accessed on 10 August 2023).

### 3.2. Substrate and Biotransformation Procedures

Pimavanserin tartrate was purchased from Shenyang Hainuowei Pharmaceutical Technology Co., Ltd. The structure was identified on the basis of its NMR spectroscopic data and its purity was determined to be 99% by HPLC analysis.

Strains were inoculated onto slant and cultured at 28 °C for 7 days to obtain the culture with abundant spores. Five mL of seed culture medium was added into the slant, then the spores were scraped and transferred into 250 mL Erlenmeyer flask containing 100 mL of seed culture medium (pH 6.5). Those flasks were incubated in a rotary shaker at 180 rpm at 28 °C for 24 h to harvest the seed culture for the next step.

To evaluate the different microbial transformation effects, growth cell transformation and resting cell transformation were, respectively performed.

For the growth cell transformation, five mL of above obtained seed culture was inoculated into the 250 mL Erlenmeyer flask containing 100 mL of bran media. After the flasks were shaken at 180 rpm at 28 °C for 48 h, 1.0 mL sterile water containing 6.0 mg pimavanserin tartrate was administered into the flask, and these flasks were culture for another 48 h. Three experimental groups were set to clarify the transformation results. A substrate transformation groups is consisted of the substrate, filamentous strains, and culture media. A substrate control groups is consisted of substrate and culture media only. And, a strain control group is consisted of strains and culture media. All the three groups are cultured equally.

For the rest cell transformation, cells with culture time of 24 h, 48 h, 72 h, 96 h, and 120 h were harvested separately by centrifugation and then washed twice with 100 mM saline to clean the media. Approximately 15 g wet weight of resting cells of each culture time age was transferred and resuspended in the 250 mL Erlenmeyer flask containing 100 mL of saline with 6.0 mg of pimavanserin tartrate. Similarly, another three groups were set, except the culture media is substitute to the sterile $H_2O$.

### 3.3. High Performance Liquid Chromatography (HPLC)

The transformation solution was extracted for three times by using 2x volume of ethyl acetate. The combined organic phase was distilled off under reduced pressure. The residue was resolved in the acetonitrile: $H_2O$ (10:90) and pass through the 0.22 μm filter before HPLC.

High performance liquid chromatography for detection was carried out using Hypersil BDS C18 (250 × 4.6 mm, 5 μm) stainless steel column. The mobile phase is consisted of aqueous phase with 0.1% formic acid—20 mmol ammonium acetate water, and organic phase with 0.1% formic acid acetonitrile (aqueous phase: organic phase: 0–10 min 80–20%; 10–30 min 80–20%; 30–35 min 20–80%; 35–40 min 20–80%; 40–50 min 80–20%). Elution was detected by UV detector at wavelength of 210 nm. Injection volume was set at 20 μL. Flow rate was adjusted to 1 mL/min and a column temperature maintained at 40 °C.

### 3.4. Optimization of the Microbial Transformation Process

Optimization for the microbial transformation process were performed by changing the substrate concentration, culture time, initial pH, culture temperature, and shaking speed. In the optimal substrate concentration experiment, the substrate concentrations were set to 0.02 mg/mL, 0.04 mg/mL, 0.06 mg/mL, 0.08 mg/mL, and 0.1 mg/mL, respectively, and the transformation rates were tested. In the experiment of optimal transformation time, The transformation time was set to 24 h, 48 h, 72 h, 96 h, and 120 h, respectively. In the optimal initial pH experiment, the pH of the culture medium was set to 5.5, 6.0, 6.5, 7.0, and 7.5, respectively. In the optimal transformation temperature experiment, the temperature of fermentation medium was set to 24 °C, 26 °C, 28 °C, 30 °C, and 32 °C, respectively. In the optimal shaking speed experiment, the shaking speed was set to 140 r/min, 160 r/min, 180 r/min, 200 r/min, and 220 r/min, respectively.

### 3.5. Preparation and Identification of Major Biotransformation Products

Chromatographic separation was performed on an ACQUITY BEH C18 column (100 mm × 2.1 mm, 1.7 μm; Waters Corp., Milford, MA, USA) using a Waters UPLC system (Waters Corp., Milford, MA, USA). The column and auto-sampler tray temperatures were kept constant at 35 °C and 10 °C, respectively. Aqueous phase was consisted of 0.1% formic acid-20 mmol/L ammonium acetate. Organic phase was consisted of 0.1% formic acid acetonitrile, gradient elution (0–4 min, 90–10%; 4–7 min, 65–35%; 7–8.5 min, 58–42%; 8.5–20 min, 50–50%; 20–23 min, 2–98%; 23–30 min, 90–10%). The mass spectrometric data were collected on an Xevo TQS mass spectrometer (Waters, USA) with a triple quadruple mass analyzer and an electrospray ionization (ESI) interface in a positive mode. The optimal parameters were set as follows: the capillary voltage, 3.0 kV; the cone voltage both for pimavanserin and IS, 30 V; the desolvation gas flow, 500 L/h; the cone gas flow, 30 L/h; the desolvation gas temperature, and 450 °C; source temperature, 110 °C. Argon was used as the collision gas, and the collision energies were 20 eV. Scanning mode: full scan, selective ion monitoring (SRM), daughter scan, and full scan range of 50–500 Da.

### 3.6. Molecular Docking

The preliminary molecular docking was performed by Modeller and AutoDock-Vina program (Modeller 9.1 and Autodock 4.2). The 5-HT$_{2A}$ receptor model was constructed by following the Bielenica's work [12]. The amino acid sequence of the 5-HT$_{2A}$ receptor is sourced from the National Center for Biotechnology Information (NCBI) in the United States as the following:

MDILCEENTSLSSTTNSLMQLNDDTRLYSNDFNSGEANTSDAFNWTVDSENRTNL-SCEGCLSPSCLSLLHLQEKNWSALLTAVVIILTIAGNILVIMAVSLEKKLQNATNYFLMSL-AIADMLLGFLVMPVSMLTILYGYRWPLPSKLCAVWIYLDVLFSTASIMHLCAISLDRYVA-IQNPIHHSRFNSRTKAFLKIIAVWTISVGISMPIPVFGLQDDSKVFKEGSCLLADDNFVLIG-SFVSFFIPLTIMVITYFLTIKSLQKEATLCVSDLGTRAKLASFSFLPQSSLSSEKLFQRSIHREP-GSYTGRRTMQSISNEQKACKVLGIVFFLFVVMWCPFFITNIMAVICKESCNEDVIGALLN-VFVWIGYLSSAVNPLVYTLFNKTYRSAFSRYIQCQYKENKKPLQLILVNTIPALAYKSSQL-QMGQKKNSKQDAKTTDNDCSMVALGKQHSEEASKDNSDGVNEKVSCV*.

## 4. Conclusions and Discussion

Pimavanserin is a drug for the treatment of Parkinson's psychosis. It is a powerful selective inhibitor of 5-hydroxytryptamine 2A (5-HT$_{2A}$) receptor, has no affinity to dopaminergic receptor, does not produce extrapyramidal effects, and will not increase the exercise burden of Parkinson's disease patients [1–3]. At present, there are few reports on the metabolism of Pimavanserin, so it is meaningful to study the valuable metabolites obtained through microbial transformation.

Compared with in vivo metabolism and chemical synthetic methods, microbial transformation distinguishes many advantages such as mild reaction conditions, regio-, and stereo-selectivity. Microbial transformation could yield novel drugs and existing drugs

could be modified so as to increase activity and decrease toxicity. Sometimes, by modification of the parent drug, its side-effects could be reduced and the stability could be increased [13]. *Cunninghamella blakesleeana* NRRL 1369 has P450 monooxygenase. Fermentation of ruscogenins (75:25, neoruscogenin-ruscogenin mixture) with *Cunninghamella blakesleeana* NRRL 1369 yielded eight previously undescribed hydroxylated compounds. Hydroxylation at C-7, C-12, C-14, and C-21 with further oxidation at C-1 and C-7 were observed with *Cunninghamella blakesleeana* NRRL 1369 [14]. *Cunninghamella blakesleeana* -mediated biotransformation of an oral contraceptive drug, levonorgestrel, was studied. Hydroxylation and dehydrogenation of levonorgestrel was observed during the bio-catalytic transformation [15]. Hydroxylated 10-deoxoartemisinins are a series of properties-improved derivatives. Via transformation of *Cunninghamella blakesleeana*, which can hydroxylate 10-deoxoartemisinin at multiple sites, the biotransformation products of 10-deoxoartemisinin have been investigated [16].

Incubation of pimavanserin with *Cunninghamella blakesleeana* AS 3.970 yielded ten unreported products (M1–M10). These transformed metabolites were identified by the various spectroscopic methods. Compared with pimavanserin, the dominant transformation product(M1) is a hydroxylated product of pimavanserin. The fragment ions are mainly *m/z* 98 and *m/z* 223, indicating that there is no structural change in the 4-fluorobenzyl and 1-methylpiperidine-4-yl parts. This is a classical transformation type of *Cunninghamella* species due to its P450 enzyme. From the other metabolites analysis, it is also could be acknowledged that multiple enzymatic reactions could be disclosed, such as single (M1/M6) and double P450 hydroxylation(M2/M7), deisobutylation (M3), defluorobenzylation (M4), deisopropyloxybenzylation (M5), methylation (M8), demethylation (M6/M7), methoxylation (M9), and defluorination (M10). These diversified reaction results also coincided that the multiple enzymatic systems in *Cunninghamella.* Further investigation could be explored if some certain metabolites could exhibit specific bioactivities.

When the molecular docking is performed, compared with the pimavanserin, two hydrogen bonds (LEU-123, PHE158) are added when it binds to 5-HT$_{2A}$ receptor, and the binding energy changes from $-9.1$ kcal/mol to $-9.3$ kcal/mol, which makes the binding force stronger. These data indicated that M1 also exhibited similar characteristics with pimavanserin, and further investigation could be performed to evaluate the bioreactivity of M1 in the near future.

**Supplementary Materials:** The following supporting information can be downloaded at: https://www.mdpi.com/article/10.3390/catal13081220/s1, Table S1. MS data and chromatographic retention time of microbial transformation metabolites of pimavanserin, Figure S1. SIR chromatograms of pimavanserin and its transformation metabolites (M1–M5), Figure S2. MS/MS spectrum of [M + H]$^+$ of M1, Figure S3. Predictive structure of M1, Figure S4. MS/MS spectrum of [M + H]$^+$ of M2, Figure S5. Predictive structure of M2, Figure S6. MS/MS spectrum of [M + H]$^+$ of M3, Figure S7. Predictive structure of M3, Figure S8. MS/MS spectrum of [M + H]$^+$ of M4, Figure S9. Predictive structure of M4, Figure S10. MS/MS spectrum of [M + H]$^+$ of M5, Figure S11. Predictive structure of M5, Figure S12. SIR chromatograms of pimavanserin and its transformation metabolites (M6–M10), Figure S13. MS/MS spectrum of [M + H]$^+$ of M6, Figure S14. Predictive structure of M6, Figure S15. MS/MS spectrum of [M + H]$^+$ of M7, Figure S16. Predictive structure of M7, Figure S17. MS/MS spectrum of [M + H]$^+$ of M8, Figure S18. Predictive structure of M8, Figure S19. MS/MS spectrum of [M + H]$^+$ of M9, Figure S20. Predictive structure of M9, Figure S21. MS/MS spectrum of [M + H]$^+$ of M10, and Figure S22. Predictive structure of M10.

**Author Contributions:** M.S.: Investigation; Q.Y.: Investigation; Y.L.: Methodology, S.C.: Methodology, X.J.: Data Curation, W.X. (Weizhuo Xu) and W.X. (Wei Xu): Resources, Supervision, and Writing—Review and Editing. All authors have read and agreed to the published version of the manuscript.

**Funding:** This research received no external funding.

**Data Availability Statement:** Data are available upon reasonable request.

**Conflicts of Interest:** The authors declare no conflict of interest.

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
