# Peer review of "Microbial Transformation of Pimavanserin by Cunninghamella blakesleeana AS 3.970"

_catalysts, doi:10.3390/catal13081220_

Round 1

Reviewer 1 Report (Previous Reviewer 1)

The sentence "Drug metabolism paves fundamental way in the development of new drug ..." should be revised to " "Drug metabolism paves the way in the fundamental development of new drug ....."

Also line 39 - this should be "Drug metabolism paves ...." as this is not an appropriate for metabolism as a whole.

The line "so microbial transformation has been proposing as a complementary in vitro model for mammalian drug metabolism" would be more accurate and grammatically correct as "so microbial transformation has been proposed as a model system for mammalian drug metabolism"

Line 45 - what do you mean by "material"? 

Line 84 - shouldn't the term be "specific transformation rate"? i.e. kinetic rate per mass.

Lines 78-90 regarding the optimal pH are repetitious and could be put into one sentence.

The names of the microbes in the methods should be in italics (lines 166-169).

"Cunningamella blakesleeana" in reference 13 should be in italics.

“M. Cyclinelloides” should be “M. circinelloides” This has not been corrected as requested in the previous version of the manuscript. There are some other changes that were not made:

The correct IUPAC name for pimavanserin is N-(4-fluorophenylmethyl)-N-(1-methylpiperidin-4-yl)-N'-(4-(2-methylpropyloxy)phenylmethyl)carbamide.

Please place a gap between the numeral and the units throughout the manuscript.  

Section 2.3: "444" should be written “444 Da” or "444 g/mol"

You have answered in the response that the control fungi Mucor and Aspergillus express CP450. Please give literature references (or access code and database to the published genes) from those species in the manuscript.

Grammar could be improved in places, but it is comprehensible as it stands.

Author Response

     Thank you very much for your comments on our manuscript entitled " Microbial transformation of Pimavanserin by Cunninghamella blakesleeana AS 3.970". These comments are encouraging and constructive, as well as very helpful to revise and improve our manuscript. After carefully studied our reviewers' comments, we have made corresponding corrections with the track mark in the manuscript. Also, we listed our answers to the reviewer's comment as below in blue italic font. Hope these efforts will make the manuscript acceptable for publication.

     The main corrections in the paper and the responses to editor’s comments are as follows:

Comments from reviewer 01

1.The sentence "Drug metabolism paves fundamental way in the development of new drug ..." should be revised to " "Drug metabolism paves the way in the fundamental development of new drug ....."  Also line 39 - this should be "Drug metabolism paves ...." as this is not an appropriate for metabolism as a whole. The line "so microbial transformation has been proposing as acomplementary in vitro model for mammalian drug metabolism" would be more accurate and grammatically correct as "so microbial transformation has been proposed as a model system for mammalian drug metabolism"

Yes, both sentences had been substituted as per our reviewer’s suggestions.

  1. Line 45 - what do you mean by "material"?

This word had been deleted for more precise expression.

3.Line 84 - shouldn't the term be "specific transformation rate"? i.e. kinetic rate per mass.

Yes, the word specific had been inserted.

4.Lines 78-90 regarding the optimal pH are repetitious and could be put into one sentence.

Yes, we’ve rephrased the sentence as “The pH value of the culture medium is closely related to microbial life activities, which. could affect the microorganism growth, metabolites accumulation, nutrients utilization, as well as the structure and activity of enzymes.”

5.The names of the microbes in the methods should be in italics (lines 166-169).

Yes, those names had been changed into italic style.

6."Cunningamella blakesleeana" in reference 13 should be in italics.

Yes, this name had been changed into italic style.

  1. “M. Cyclinelloides” should be “M. circinelloides” This has not been corrected as requested in the previous version of the manuscript.

Sorry for our careless, this name had been corrected in Line 69.

8.The correct IUPAC name for pimavanserin is N-(4- fluorophenylmethyl)-N-(1-methylpiperidin-4-yl)-N'-(4-(2- methylpropyloxy)phenylmethyl)carbamide. Please place a gap between the numeral and the units throughout the manuscript.

Yes, this IUPAC name and other names had been modified.

9.Section 2.3: "444" should be written “444 Da” or "444 g/mol"

Yes, it had been changed into “444 Da”.

10.You have answered in the response that the control fungi Mucor and Aspergillus express CP450. Please give literature references (or access code and database to the published genes) from those species in the manuscript.

Yes, these CYP450 genes could found from CYtochrome P450 Engineering Database from BioCatNet (https://cyped.biocatnet.de/). This had been supplemented to the Material and Methods section in lines 231-233.

Reviewer 2 Report (Previous Reviewer 3)

The manuscript titled 'Microbial transformation of Pimavanserin by Cunninghamella blakesleeana AS 3.970' attempted to study the pimavanserin metabolism profiles in 14 vitro using eight strains of fungi.

The manuscript appears to be sound with only minor technical and language errors. The results are generally well presented except for Figure 5 (Molecular docking results of 5-HT2A receptor with pimavanserin(A) and M1(B)) which was not mentioned or mistakenly mentioned as Figure 4 in the text. The results of this paper do meet the impact and innovation criteria of this journal. 

Author Response

     Thank you very much for your comments on our manuscript entitled " Microbial transformation of Pimavanserin by Cunninghamella blakesleeana AS 3.970". These comments are encouraging and constructive, as well as very helpful to revise and improve our manuscript. After carefully studied our reviewers' comments, we have made corresponding corrections with the track mark in the manuscript. Also, we listed our answers to the reviewer's comment as below in blue italic font. Hope these efforts will make the manuscript acceptable for publication.

     The main corrections in the paper and the responses to editor’s comments are as follows:

Comments from reviewer 02

The manuscript appears to be sound with only minor technical and language errors. The results are generally well presented except for Figure 5 (Molecular docking results of 5-HT2A receptor with pimavanserin(A) and M1(B)) which was not mentioned or mistakenly mentioned as Figure 4 in the text. The results of this paper do meet the impact and innovation criteria of this journal.

Thanks for our reviewer, Figure 4 had been changed to Figure 5.

Reviewer 3 Report (Previous Reviewer 4)

This manuscirpt can be accepted now.

Author Response

  Thank you very much for your reviewing our manuscript entitled " Microbial transformation of Pimavanserin by Cunninghamella blakesleeana AS 3.970".

           We are happy to know your consent for our publication.

           Thanks.

Reviewer 4 Report (New Reviewer)

Dear authors, thank you for your work that is of interest to the journal.

I believe it is usually well written and that results are supported by the data. Also the material and methods section is detailled enough dor someone in the field to try reproduce your work.

Therefore i will recommend it to the editor for publication in the catalysts journal and in the special issue with minor revision.

I have one particular concern in the introduction it is said that "eight strains of fungi are used to study the pimavanserin metabolism". However as it is stated this molecule is used to treat parkinson disease in humans. Also i cannot understand the link between your study and the use of fungi. Maybe you could have tried to identify degradation products originating from pimavanserin directly in patients blood samples?

I suggest that at the end of the introduction you provide a sentence stating clearly what is the aim of the study and why fungi were choosen in particular
With my best regards

the language is generally good and understandable. It may require a final revision for typos.

Author Response

     Thank you very much for your comments on our manuscript entitled " Microbial transformation of Pimavanserin by Cunninghamella blakesleeana AS 3.970". These comments are encouraging and constructive, as well as very helpful to revise and improve our manuscript. After carefully studied our reviewers' comments, we have made corresponding corrections with the track mark in the manuscript. Also, we listed our answers to the reviewer's comment as below in blue italic font. Hope these efforts will make the manuscript acceptable for publication.

     The main corrections in the paper and the responses to editor’s comments are as follows:

Comments from reviewer 04

Therefore i will recommend it to the editor for publication in the catalysts journal and in the special issue with minor revision.

I have one particular concern in the introduction it is said that "eight strains of fungi are used to study the pimavanserin metabolism". However as it is stated this molecule is used to treat parkinson disease in humans. Also i cannot understand the link between your study and the use of fungi. Maybe you could have tried to identify degradation products originating from pimavanserin directly in patients blood samples? I suggest that at the end of the introduction you provide a sentence stating clearly what is the aim of the study and why fungi were choosen in particular.

Thanks for our reviewer’s suggestions. Currently, there are few reports on the metabolism of Pimavanserin in human body/in vivo, so it is valuable to study the diversified metabolites obtained through microbial transformation. After fungal transformation, it is possible to obtain some new and promising substances with better efficacies, such as reducing toxic effects and improving therapeutic effects.

So, we’ve supplemented the words “in vivo” in line 53 to emphasis that the microbial transformation is a nice alternation/supplementation for human body metabolism for pimavanserin.

This manuscript is a resubmission of an earlier submission. The following is a list of the peer review reports and author responses from that submission.

Round 1

Reviewer 2 Report

This manuscript intends to investigate the metabolism of pimavanserin by microbial transformation with the help of several fungi species. Microbial transformation is widely accepted as a useful in vitro model to study drug metabolism in mammalian cells, although mainly as a complementary and exploratory technique.

First of all, English language and style must be extensively edited. Some parts of the manuscritp are very difficult to understand. Also, the methods are not adequately explained.

In general, the work described here lacks depth, as well as clarity. It is unclear why those specific fungi strains have been selected. Also, the selection of C. blakesleana AS 3.970 seems to be based in a rather preliminary experiment, for which the authors do not explain how the culture conditions were chosen. The usefulness of the presented optimization of the culture conditions is doubtful, and seems to be based only in global transformation percentages, without taking into account potential differences in metabolic routes.

The identificacion of transformation products is interesting, but again the usefulness of the results is debatable, apart from a general chemical analysis point of view. More work would be needed to confirm if these compounds could also appear when pimavanserin is metabolised by mammalian cells. Besides, more attention should be paid to the function of the produced metabolites.

In summary, this manuscript is not suitable for publication in Catalysts.

Reviewer 3 Report

The manuscript appears to be sound with only minor technical errors. The results are generally well presented. But in my opinion, the results of this paper do meet the impact and innovation criteria of this journal.

Reviewer 4 Report

This work focus on Pimavanserin’s microbial transformation in fungal hosts, with one major metabolites M1 verified, proved to be Pimavanserin’s hydroxylated derivatives, and its molecular docking data shows potential against Parkinson's disease. While the rest 9 metabolites remain to be proposed structure, thus a great number of transformation information was unknown, as well as their pharmacological/toxicological potential. To sum up, this job offered insufficient statistics in investigating Pimavanserin’s microbial transformation.

Some suggestions were given as follows:

1.      Section 1, paragraph 1, the last sentence, “…and functional activity at 5-HT2B receptors…”, 2B should be subscript.

2.      Figure 1, “A” is missed in the first HPLC spectrum, and 26.21 showed twice in scheme B. The HPLC spectrums should be annotated for better understanding.

3.      Table 2, footnote b, MHZ should be corrected as MHz.

4.      Section 4, paragraph 2, It’s inappropriate for the authors to claim that “These transformed metabolites were identified by the various spectroscopic methods”, according to the results, compounds M2-M9 were only scanned by LC-MS/MS, and yet it’s not enough to be identified for these structures. The authors should display more information to verify M2-M10.

5.      Author Contributions, “…Sulan Cai1(Methodology)…”, there is an unwanted 1.